# Medical Cannabis Certification in a Large Pediatric Oncology Center

**DOI:** 10.3390/children6060079

**Published:** 2019-06-17

**Authors:** Mary M. Skrypek, Bruce C. Bostrom, Anne E. Bendel

**Affiliations:** Cancer and Blood Disorders Program, Division of Pediatric Hematology-Oncology, Children’s Minnesota, 2530 Chicago Avenue South, CSC-175, Minneapolis, MN 55404, USA; Maggie.Skrypek@childrensmn.org (M.M.S.); Anne.Bendel@childrensmn.org (A.E.B.)

**Keywords:** medical cannabis, medical marijuana, pediatric oncology, chemotherapy-induced nausea, cancer pain, end-of-life care

## Abstract

In Minnesota, medical cannabis was approved for use in 2014. From July 2015 to February 2019, our center certified 103 pediatric and young adult patients for the use of medical cannabis under the qualifying conditions of cancer and treatment-related symptoms. Here, we provide a review of the literature on medical cannabis use in pediatric and young adult cancer patients. We also provide demographic data on our patients certified for medical cannabis. The most common diagnoses were leukemia/lymphoma (36%), brain tumors (37%), and malignant solid tumors (26%). The most common indications were chemotherapy-related nausea, pain, and cancer cachexia. The age range at certification was 1.4–28.7 years (median 15.3 years). The time from cancer diagnosis to certification ranged from 0.5–197 months (median 8.9 months). The majority (94%) were certified during their first line of treatment. In the 32 patients who died from recurrent or progressive cancer, the time from certification to death was 1.3–30.3 months (median 4.4 years). Despite requesting certification, a subset (24%) never had medical cannabis dispensed. In our experience, pediatric and young adult oncology patients are interested in medical cannabis to help manage treatment-related symptoms. Ongoing analysis of this data will identify the therapeutic efficacy of medical cannabis.

## 1. Introduction

In August of 2014, the state of Minnesota passed legislation (Minn. Stat. Sec. §§152.22 to 152.37) providing legal protection for the use of medical cannabis products in a small subset of patients with certain debilitating medical conditions, with the recommendation of a licensed healthcare practitioner. The Minnesota Medical Cannabis Act also allowed for designation of a caregiver for patients who are developmentally unable, due to age or disability, to obtain or self-administer medication, and therefore made medical cannabis accessible to the pediatric population. The legislation mandated that there be a registry program managed by the Minnesota Department of Health (MDH), to allow for close monitoring and to ensure appropriate use of medical cannabis. This registry has allowed for further breakdown analyses of access and use based on demographic information and qualifying medical conditions. In June of 2015, registry applications were opened, and in July of 2015, medical cannabis products were first distributed to those patients on the state registry. In its original form, the Minnesota Medical Act’s qualifying conditions included cancer and terminal illness with life expectancy of under one year (with specific sub-qualifying conditions related to the diagnosis or treatment, including “severe or chronic pain”, “nausea or severe vomiting”, and “cachexia or severe wasting”) as well as glaucoma, human immunodeficiency virus/acquired immunodeficiency syndrome (HIV/AIDS), Tourette Syndrome, amyotrophic lateral sclerosis (ALS), seizures, severe or persistent muscle spasms, and inflammatory bowel disease. Since then, there have been addendums approving additional qualifying conditions, which include autism spectrum disorder, post-traumatic stress disorder (PTSD), and obstructive sleep apnea (OSA) [1,2]. The MDH reports a continued rise in certification and enrollment in the Minnesota medical cannabis program since its initiation. Notably, physicians must first certify patients who meet the above qualifying and sub-qualifying conditions for medical cannabis use before the patients can enroll in the Minnesota Medical Cannabis Registry. Enrollment includes an annual fee of $200.00 plus the cost of the medical cannabis.

With a continued rise in qualifying conditions and a subsequent rise in the prevalence of use of medical cannabis, particularly in the setting of ongoing national dialogue about the decriminalization of cannabis, it is increasingly common for the pediatrician to be asked about the use of medical cannabis. For the pediatric oncologist, questions pertaining to potential risks and benefits of medical cannabis for the management of cancer or treatment-related side-effects, as well as the potential role of medical cannabis as an anti-cancer treatment have become an expected part of discussions with patients and their families. To date, there is limited data for evidence-based use of medical cannabis in the pediatric population, and these discussions are most commonly facilitated by information obtained through small patient cohort experiences or anecdotal reports.

The majority of publications on the subject of medical marijuana and its use in patients with cancer are adult-based. Boehnke et al. analyzed state registry data to further characterize the conditions for which adults were being certified, and compared this to published evidence for the efficacy of medical cannabis by the National Academics of Sciences, Engineering, and Medicine. They report that 85.5% of patient-reported qualifying conditions had either substantial or conclusive evidence of therapeutic efficacy. They also report that the leading qualifying conditions used are those of chronic pain, multiple sclerosis with spasticity, and chemotherapy-induced nausea [3]. A very early publication from 1994 using recreational marijuana in patients with chemotherapy-induced nausea found benefit in 50% of patients [4]. More recent publications in adults with cancer have shown significant benefits of medical cannabis and recreational marijuana, with few and manageable side effects [5,6].

The American Academy of Pediatrics (AAP)’s current policy opposes medical cannabis use in children and young adults. However, the policy states that ‘The AAP recognizes that marijuana may currently be an option for cannabinoid administration for children with life-limiting or severely debilitating conditions and for whom current therapies are inadequate.’ This seemingly provides some leniency within their stance on use for patients with cancer [7]. Despite mounting evidence based on adult studies as well as rising interest in and use of medical cannabis in the pediatric population, it remains controversial.

To date, there is only one publication describing the use of medical cannabis in a pediatric oncology population. In the Journal of Pediatrics, Ananth et al. published their findings from a multicenter cross-sectional survey characterizing pediatric provider practices, knowledge, attitudes, and barriers related to medical marijuana and its use in the care model of pediatric patients with cancer [8]. In this article, the authors infer ongoing interest and demand from their patients and their caregivers for access to medical marijuana. The authors identify potential uses for medical marijuana as: Antiemetic effects, appetite stimulation, pain management, seizure therapy, and antineoplastic potential. They also identify potential risks of medical marijuana, including long-term neurocognitive impacts on the developing brain and negative mental health sequelae. Based on studies by the National Institute on Drug Abuse, they conclude that medical cannabis may ameliorate nausea. They also conclude that it may have a major positive impact on the experience of pain. Interestingly, this study found that certifying providers demonstrate ongoing apprehension about its use, with a bias for use in patients with advanced stages of disease or in efforts toward palliation. Ultimately, the authors conclude that in the face of insufficient data, pediatric providers should exercise caution in recommending medical marijuana. However, there is data supporting the efficacy of dronabinol and nabilone, two Federal and Drug Administration (FDA) approved synthetic cannabinoids, for the treatment of chemotherapy-induced nausea and vomiting in children, which suggests that medical cannabis may also be an effective anti-emetic for children undergoing cancer treatment [9,10,11,12].

## 2. Materials and Methods

Records from the certified medical cannabis providers in the Division of Pediatric Hematology–Oncology at Children’s Minnesota from July 2015 to February 2019 were obtained from the Minnesota Medical Cannabis Registry website [1]. Data extracted included name, date of medical cannabis certification, indication for medical cannabis certification, and if medical cannabis was dispensed. Patients were included if they had a cancer diagnosis. From the Children’s Minnesota Cancer and Blood Disorders database, the following data were extracted: Diagnosis, date of diagnosis, date of relapse or tumor progression, and date of death. Diagnostic sub-groups were identified as follows: Brain Tumors, low and high-grade; Leukemia; Lymphoma; Malignant Solid Tumors; and Mast Cell Activation Syndrome, a chronic hematologic disorder. The specific diagnoses are listed in the Appendix A.

Statistical analysis and graphing were done by SPSS version 23. For time variables, the mean, median, and percentile distribution were calculated. Analysis for difference between diagnostic groups and other time variables were computed by chi-square analysis, with Kruskal–Wallis test use for non-parametric comparisons. Bonrerroni correction for multiple analyses was not included due to the highly significant *p* values. This study was approved by the Children’s Minnesota Institutional Review Board.

## 3. Results

Herein we report demographic information including: Age of certification, gender, timing of certification, cancer diagnosis, indication for certification, and if medical cannabis was dispensed.

We have certified 101 patients for the use of medical cannabis for cancer and treatment-related symptom management. The median age at certification was 15.3 years, with a range of 1.4 to 28.7 years (Figure 1). Twenty percent of the patients were under 6 years of age, with 37 percent under 12 years of age. There were 57 males (56%) and 44 females (44%). There were no significant differences by age at certification, time from diagnosis to certification, or time from certification to death by gender. There were no significant differences by gender in the likelihood that medical cannabis was dispensed or certification in regards to disease progression.

The median time from oncologic diagnosis to medical cannabis certification was 8.9 months, with a range from 0.5 to 197 months (Figure 2). There was no significant difference in time from diagnosis to certification by diagnostic group. Ninety-four percent of the patients were certified during their first line of treatment (Table 1), although patients with brain tumors tended to have higher certification after progression compared to other diagnoses (chi-square *p* value = 0.054). The median time from certification to death, which occurred in 29 patients (29%), was 4.4 months, with a range of 1.3 to 30.3 months. There was no significant difference in time from certification to death by diagnostic group. There was no significant difference in time from diagnosis to certification for the patients who died or survived. Overall, 77 of 101 patients (76%) were dispensed medical cannabis (Figure 3). There were no significant differences by chi-squared analysis on percent dispensed by age or diagnostic group.

A box plot of the age at certification by diagnostic group is shown in Figure 4. The age distribution was significantly different by non-parametric analysis using the Kruskal–Wallis test (*p* < 0.001). Combining the diagnoses into three diagnostic sub-groups (all brain tumors, all leukemia and lymphoma, and all solid tumors) showed significant differences between all three groups by independent samples t-test, with *p*-values of less than or equal to 0.002 for all three comparisons. The median, mean, and standard deviation of age at certification in years were: 8.7 and 9.1 ± 5.7 for brain tumors, 15.9 and 14.2 ± 5.7 for solid tumors, and 18.1 and 17.2 ± 5.3 for leukemia and lymphoma. Table 2 demonstrates the primary reason for medical cannabis certification by diagnostic group. The distributions were significantly different among the diagnostic groups (*p* = 0.002 by chi-square). The primary reason for certification was nausea in patients with leukemia, lymphoma, and high-grade brain tumors, whereas it was pain in solid tumors. Table 3 lists the secondary indications for medical cannabis by diagnostic group. The distributions were significantly different among the diagnostic groups (*p* < 0.001 by chi-square). Only patients with brain tumors showed interest in medical cannabis for possible anti-tumor effects. The term “end-of-life” was only used in patients with solid tumors.

## 4. Discussion

This report describes our nearly four-year experience with certifying pediatric patients with a diagnosis of cancer for the use of medical cannabis. In our experience, patients and caregivers are pursuing the use of medical cannabis in a wide variety of cancer diagnoses to aid in the management of treatment-related symptoms. The primary qualifying condition for which families sought certification for medical cannabis was chemotherapy-induced nausea, followed by intractable pain. Nausea is also a common secondary indication along with cachexia. It has been our philosophy to utilize medical cannabis early following a diagnosis requiring the use of chemotherapy. Our data confirms this practice, with 94% of our patients undergoing certification for medical cannabis use during their first line of treatment. It has been our experience that traditional anti-emetics (ondansetron, granisetron, aprepitant, lorazapam, scopolamine, and dexamethasone [solid tumors only]) often provide suboptimal control of nausea and vomiting. It has also been our experience that medical cannabis, when added to these traditional anti-emetic regimens, improves the patient chemotherapy experience and quality of life.

The mean age at time of certification for our patients was 15.3 years. This varied significantly based on the diagnostic sub-group, noting that the mean age for certification of patients with brain tumors was much younger (8.7 years) in comparison to leukemia and lymphoma patients, who were much older (18.1 years). The younger age seen in the brain tumor group is due in part to the highly emetogenic chemotherapy used for the treatment of central nervous syndrome (CNS) embryonal tumors, which are typically seen in infants and young children. We extrapolate that this age difference may also be accounted for by the hope of parents that medical cannabis might provide anti-tumor effects, given the poorer overall survival for patients with brain tumors compared to those with a diagnosis of leukemia or lymphoma, and therefore willingness to utilize interventions with less data to support its use and therapeutic efficacy. Interestingly, although not a qualifying condition, hope for anti-tumor effect was noted in the brain tumor population, ranking second behind nausea as the most common indications for the use of medical cannabis. Recent reviews of preclinical research suggest the possibility of an effect by tetrahydrocannabinol (THC) and cannabidiol (CBD) against various CNS tumors, however, the authors caution that more research is needed [6,13]. We have also seen that young children with brain tumors experience a decrease in irritability while on medical cannabis, which may also account for its use in younger patients. In regards to the relatively older ages of leukemia patients certified for medical cannabis, it has been our observation that chemotherapy compliance for teenagers and young adults, particularly leukemia patients, improves with the addition of medical cannabis as they are not as burdened by life-interfering chemotherapy side effects. Therefore, we routinely recommend medical cannabis for symptoms management in this age group.

All of our patients who were certified for medical cannabis had expressed an interest in this treatment, but only 76% had medical cannabis dispensed. We currently do not have data regarding the reason why some patients did not follow through on enrolling in the Minnesota medical cannabis registry and/or pick up their medical cannabis prescription, but the expense of medical cannabis ($200.00 annual fee and an additional fee for each dispensed prescription, both of which are not covered by medical insurance) might be the etiology.

Our experience with the use of medical cannabis in children and young adults has been very positive. We assert that ongoing analysis and publishing of this data needs to occur to better understand the trends in the use of medical cannabis and to further characterize the safety and efficacy of medical cannabis in pediatric and young adult cancer patients. Our hope is that our anecdotal experience reported herein, bolstered by a more detailed analysis of our patients’ outcomes in a future publication, will support the benefit of medical cannabis for the management of chemotherapy-related symptoms in children and young adults, and thus encourage pediatric oncologists to access medical cannabis for symptom management early in the cancer diagnosis and not just for end-of-life care.

## List of Abbreviations

AbbreviationFull NameTHCTetrahydrocannabinolCBDCannabidiol

## Figures and Tables

**Figure 1 children-06-00079-f001:**
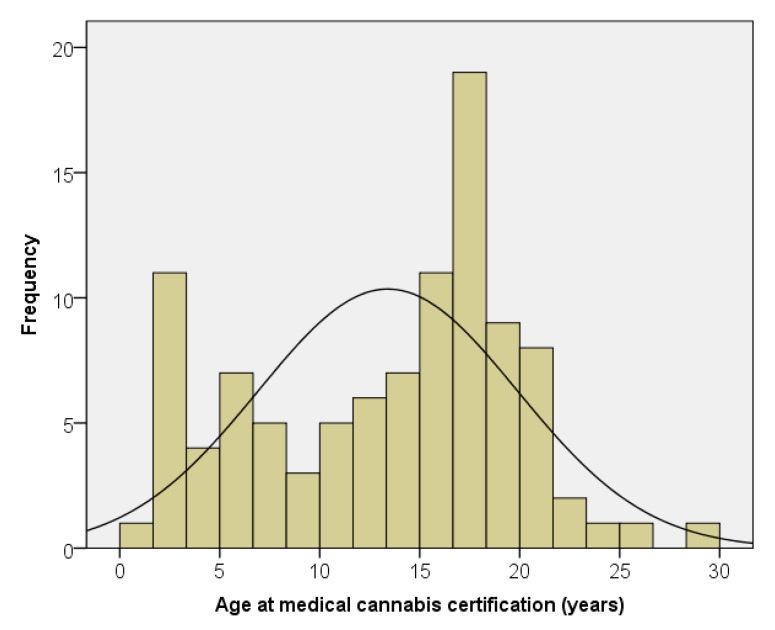
Age at medical cannabis certification.

**Figure 2 children-06-00079-f002:**
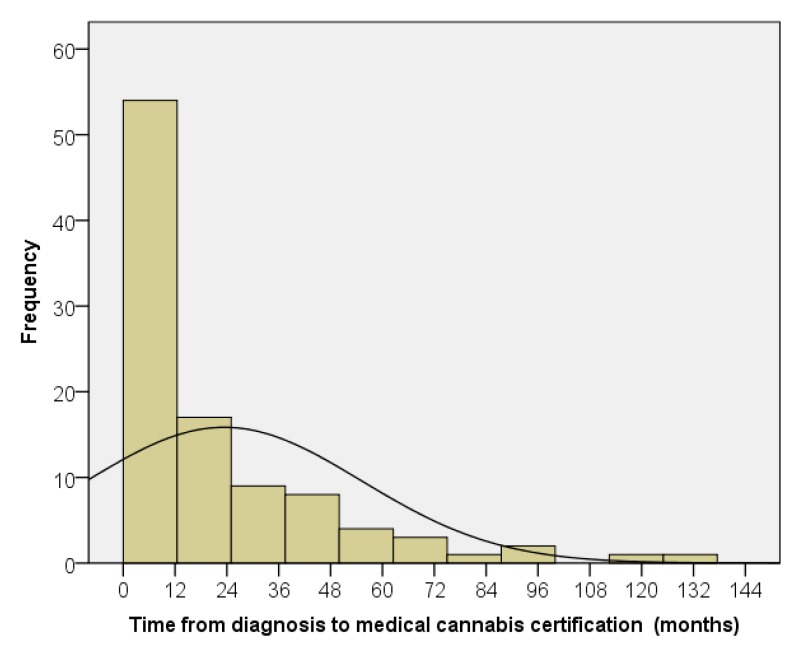
Time from cancer diagnosis to medical cannabis certification.

**Figure 3 children-06-00079-f003:**
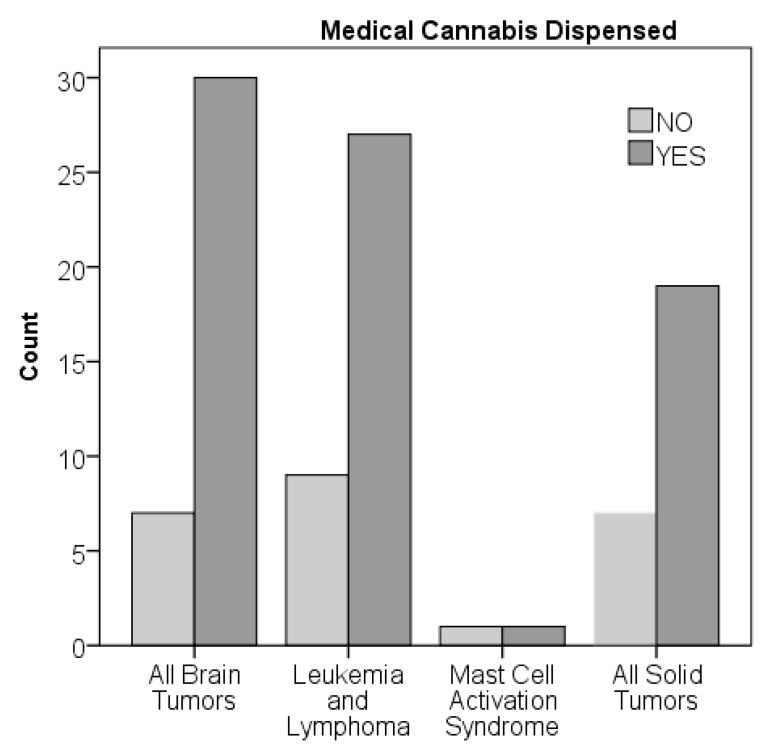
Medical cannabis dispensed by diagnostic group.

**Figure 4 children-06-00079-f004:**
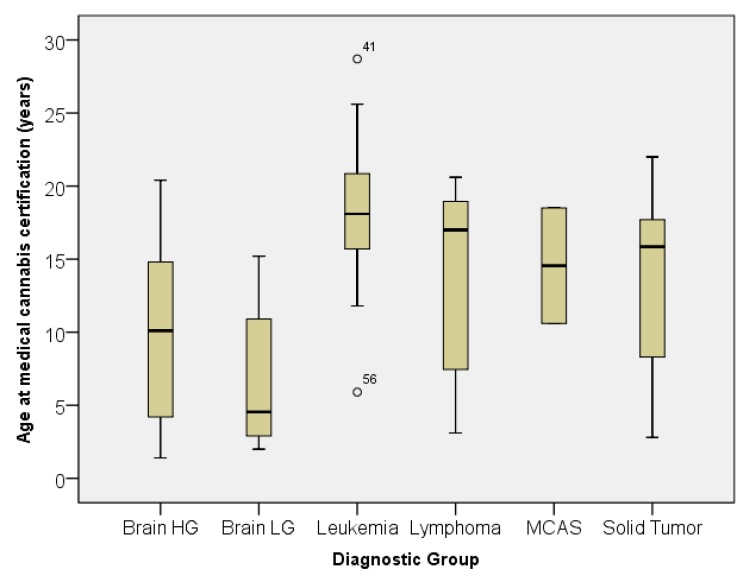
Age at medical cannabis certification by diagnostic group. Figure 4 caption: Box plot of age in years at the time of medical cannabis certification by diagnostic group. Outliers are denoted by a circle with unique patient number (UPN). Brain LG = Brain Tumor low-grade. Brain HG = Brain Tumor high-grade. MCAS = Mast Cell Activation Syndrome.

**Table 1 children-06-00079-t001:** Certification relative to disease progression.

Diagnostic Group	After	Before	No Progression	TOTAL
Brain Tumors	5	16	16	37
Leukemia & Lymphoma	0	8	28	36
Solid Tumors	1	10	15	26
Mast Cell Activation	0	1	1	2
TOTALS	6	35	60	101

N/A = Not Applicable, as there was no progression or relapse.

**Table 2 children-06-00079-t002:** Primary reason for medical cannabis certification by diagnostic group.

Diagnostic Group	Nausea	Pain	Seizure	Autism	Cachexia	TOTAL
Brain Tumor Low-Grade	4	2	2	0	0	8
Brain Tumor High-Grade	20	8	0	0	2	30
Leukemia	25	2	1	0	0	28
Lymphoma	6	2	0	0	0	8
Solid Tumor	9	15	0	1	0	25
Mast Cell Activation	0	2	0	0	0	2
TOTAL	64	31	3	1	2	101

**Table 3 children-06-00079-t003:** Secondary indications for medical cannabis by diagnostic group.

Diagnosis	Cachexia	Anti-tumor	Nausea	Pain	End of Life	Seizure	Autism	TOTAL
Brain Tumor Low-Grade	1	3	1	3	0	0	0	8
Brain Tumor High-Grade	10	16	3	1	1	1	0	32
Leukemia	6	0	0	4	0	0	2	12
Lymphoma	1	0	2	0	0	0	0	3
Solid Tumor	5	0	8	0	6	0	0	19
Mast Cell Activation	0	0	0	0	0	0	0	0
TOTAL	23	19	14	8	7	1	2	74

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
