# Peer review of "Medical Cannabis Certification in a Large Pediatric Oncology Center"

_children, 2019, doi:10.3390/children6060079_

Round 1

Reviewer 1 Report

Thank you for the opportunity to review manuscript children-497290 entitled “Medical cannabis certification in a large pediatric oncology center”.

The authors report demographic data on patients in a pediatric oncology center certified for the use of medical cannabis over a period of four years. They report significant differences between six diagnostic groups regarding age at certification and indication. Then, the authors identify nausea, pain, and hope for an anti-tumor effect as the main reasons for certification in their population. This retrospective analysis is embedded in a brief literature review alongside with controversial aspects and the authors’ positive clinical experience in the discussion section.

These are my remarks in detail:

1)    Introduction:

a)    The authors lead into the subject with a controversial discussion and a small literature review. The clearly state their intention for the report opposing demonization of cannabis use. While this is unusual for an introduction section – one would expect it in the discussion – I personally find it refreshing but the reader might get the impression of a tendentious report. Maybe a clearer structure of the literature review line 56-79 after the start in line 53-56 can help.

b)    Line 79-81: The authors touch the field of barriers to medical cannabis very shortly. Since this is a large area of discussion with even political dimensions, I suggest omitting it here in this context.

c)    Line 82-85: Here, the reader may benefit from a clearer grammar. To me it is not entirely clear whether these sentences belong to the literature review.

2)    Materials an Methods:

a)    Line 90: Website references should have the date accessed included. “Data abstracted” should read “Data extracted” throughout in my opinion.

b)    Line 91: “Hemonc” is clinical slang to me and should be revised.

c)    Line 93: Please state who established the patient groups and on what basis. Are these patients in first line therapy or relapsed?

d)    Line 95-97: The authors state what data they did not retrieve and point to a future publication. This belongs into a section on “Limitations” of the study.

e)    Line 100: SPSS version should be stated. It becomes clear that the results were graphed in histogram and box plots, I think there is no need to state it.

f)     Line 103: I wonder if the statistical analysis includes a Bonferroni correction for multiple analysis, please clarify. I would suggest to adjust for multiple analysis.

3)    Results:

a)    Line 106: “Herein we report…”

b)    Line 109-116: The authors report on figure1-3. I think adding the median and IQR to the graphs makes it easier for the reader.

c)    Line 117-123 and figure 4: Do the bold lines in the boxes refer to the median or to the mean? In the text, the authors report the mean. I may be a good idea to state why they used what.

d)    Line 123-130 and tables 1-2: The tables might benefit from an extra column with the p value.

4)    Discussion:

a)    In my opinion, it is worthwhile discussing the results a bit more in detail. The authors report their clinical philosophy, but first the results at hand should be discussed. One may ask several interesting questions:

i)     How can the age at cannabis certification in the respective diagnostic groups be interpreted? The average leukemia patient is much younger, patients with high grade gliomas are usually a bit older, for solid tumors it might fit (if the group comprises pts. with osteosarcomas/Ewing tumors etc.)

ii)   How does the above-mentioned point relate to time from diagnosis to certification? The authors state their philosophy in line 161 but it is not clear how many patients with relapsed tumors are in the population or if the certification occurs in first line therapy.

iii)  How do the above-mentioned points relate to time to death?

b)    Line 157-160: The authors discuss nausea before and after this part, I suggest discussing one aspect and then the next.

c)    Line 161-170: The authors report their clinical experience, which is a valuable resource. However, this is not related to the results at hand and should clearly be separated and preceded by the discussion of the study’s results.

d)    Line 171-175: The barriers to cannabis use is not part of the data. Only the results at hand should be discussed.

e)    Line 175: The authors summarize a positive experience, which is not part of the data. I recommend summarizing the findings at hand and express their hopes and future directions in an extra paragraph.

5)    Abstract:

a)    The abstract should be re-written and adjusted to the results and the interpretation of the data at hand.

Author Response

Reviewer 1

Comments to the Author:

The authors report demographic data on patients in a pediatric oncology center certified for the use of medical cannabis over a period of four years. They report significant differences between six diagnostic groups regarding age at certification and indication. Then, the authors identify nausea, pain, and hope for an anti-tumor effect as the main reasons for certification in their population. This retrospective analysis is embedded in a brief literature review alongside with controversial aspects and the authors’ positive clinical experience in the discussion section.

These are my remarks in detail:
1)    Introduction:

a)    The authors lead into the subject with a controversial discussion and a small literature review. The clearly state their intention for the report opposing demonization of cannabis use. While this is unusual for an introduction section – one would expect it in the discussion – I personally find it refreshing but the reader might get the impression of a tendentious report. Maybe a clearer structure of the literature review line 56-79 after the start in line 53-56 can help.

RESPONSE: We have removed the controversial discussion from this section and have added background information on the Minnesota Medical Cannabis Registry and a review of medical cannabis use in the pediatric oncology population.

b)    Line 79-81: The authors touch the field of barriers to medical cannabis very shortly. Since this is a large area of discussion with even political dimensions, I suggest omitting it here in this context.

RESPONSE: We omitted the discussion on barriers to medical cannabis.

c)    Line 82-85: Here, the reader may benefit from a clearer grammar. To me it is not entirely clear whether these sentences belong to the literature review.

RESPONSE: We omitted these sentences because they no longer pertained with the updated literature review.

2)    Materials and Methods:

a)    Line 90: Website references should have the date accessed included. “Data abstracted” should read “Data extracted” throughout in my opinion.

RESPONSE: We exchanged “Data abstracted” with “Data extracted”.

b)    Line 91: “Hemonc” is clinical slang to me and should be revised.

RESPONSE: We exchanged “Hemonc” with “Hematology-Oncology”.

c)    Line 93: Please state who established the patient groups and on what basis. Are these patients in first line therapy or relapsed?

RESPONSE: We reviewed the diagnoses of the oncology patients whom we certified for medical cannabis. We sub-grouped the cancers into Brain Tumors (low grade and high grade), Solid Tumors, Leukemia, Lymphoma and Mast Cell Activation Syndromes.  This sub-grouping of pediatric cancers is common because patients within each subgroups typically receive similar treatments and have similar outcomes.  We did not review data on if these patients are in first line therapy or if they have relapsed.  We agree that this is important information and it will be included in our subsequent publication currently in the process of consenting patients to obtain detailed information.     

d)    Line 95-97: The authors state what data they did not retrieve and point to a future publication. This belongs into a section on “Limitations” of the study.

RESPONSE: We removed these limitations from this section.

e)    Line 100: SPSS version should be stated. It becomes clear that the results were graphed in histogram and box plots, I think there is no need to state it.

RESPONSE: We added SPSS version and removed the statement that results were graphed in histogram and box plots.

f)     Line 103: I wonder if the statistical analysis includes a Bonferroni correction for multiple analysis, please clarify. I would suggest to adjust for multiple analysis.

RESPONSE:  Due to the highly significant p-values we do not feel Bonferroni correction is necessary as it would be impossible with the limited number of comparisons we did for the corrected p-value to become insignificant

3)    Results:

a)    Line 106: “Herein we report…”

RESPONSE: We added “we” to “Herein we report….”

b)    Line 109-116: The authors report on figure1-3. I think adding the median and IQR to the graphs makes it easier for the reader.

RESPONSE: A boxplot is a graph that gives you a good indication of how the values in the data are spread out. ... Boxplots are a standardized way of displaying the distribution of data based on a five number summary (“minimum”, first quartile (Q1), median, third quartile (Q3), and “maximum”).

c)    Line 117-123 and figure 4: Do the bold lines in the boxes refer to the median or to the mean? In the text, the authors report the mean. I may be a good idea to state why they used what.

RESPONSE: See above.  We also added median to text to clarify

d)    Line 123-130 and tables 1-2: The tables might benefit from an extra column with the p value.

RESPONSE: We feel that having the values in the text is sufficient

4)    Discussion:

a)    In my opinion, it is worthwhile discussing the results a bit more in detail. The authors report their clinical philosophy, but first the results at hand should be discussed. One may ask several interesting questions:

i)     How can the age at cannabis certification in the respective diagnostic groups be interpreted? The average leukemia patient is much younger, patients with high grade gliomas are usually a bit older, for solid tumors it might fit (if the group comprises pts. with osteosarcomas/Ewing tumors etc.)

RESPONSE: We addressed why we believe the the brain tumor population was younger than the leukemia/lymphoma population.  We feel if is due to a) many of the patients have CNS embryonal tumor which occur in infants and young children and the chemotherapy they receive is highly emetogenic; b) pediatric brain tumors have a poorer survival than leukemia and lymphoma and parents are often looking for any well tolerated alternative treatment that might have anti-tumor properties; and anecdotal data suggests that medical cannabis can help with irritability seen in young brain tumor patients.

ii)   How does the above-mentioned point relate to time from diagnosis to certification? The authors state their philosophy in line 161 but it is not clear how many patients with relapsed tumors are in the population or if the certification occurs in first line therapy.

RESPONSE: This data was not assessed at this time and will be included in next publication

iii)  How do the above-mentioned points relate to time to death?

RESPONSE: This data was not assessed at this time and will be included in next publication.

b)    Line 157-160: The authors discuss nausea before and after this part, I suggest discussing one aspect and then the next.

        RESPONSE: We discussed nausea in only one section of the discussion.

c)    Line 161-170: The authors report their clinical experience, which is a valuable resource. However, this is not related to the results at hand and should clearly be separated and preceded by the discussion of the study’s results.

RESPONSE: We discussed the results first and then added our clinical experience.

d)    Line 171-175: The barriers to cannabis use is not part of the data. Only the results at hand should be discussed.

RESPONSE: We removed the comments on barriers to cannabis.

e)    Line 175: The authors summarize a positive experience, which is not part of the data. I recommend summarizing the findings at hand and express their hopes and future directions in an extra paragraph.

RESPONSE: We edited the conclusions stating more analysis was needed to determine the efficacy of medical cannabis in pediatric and young adult oncology patients.

5)    Abstract:

a)    The abstract should be re-written and adjusted to the results and the interpretation of the data at hand.

RESPONSE: The abstract was rewritten to only include the results of our analysis.

Reviewer 2 Report

As important as this topic is, my bottom line conclusion is that this manuscript is simply too preliminary. It may be important to know for whom cannabis has been recommended, but without data on which cancer patients received it, how they benefitted, and what adverse events were associated, there is simply not enough information to merit publication. I strongly encourage the authors to try again when a complete data set is available. 

Additional comments:

1) One mention of "marijuana" is sufficient. The proper name is cannabis and "marijuana" has no place in modern scientific discourse.

2) The literature review must be much more intensive and take into account the vast range of available publications. One example is particularly important: 

Abrahamov, A., and R. Mechoulam. 1995. "An efficient new cannabinoid antiemetic in pediatric oncology." Life Sci 56 (23-24): 2097-102.

This type of data showing marked benefit of delta-8-THC on chemotherapy-associated nausea without associated adverse events would greatly bolster the authors' observations.

Another: 

Duran, M., E. Perez, S. Abanades, X. Vidal, C. Saura, M. Majem, E. Arriola, M. Rabanal, A. Pastor, M. Farre, N. Rams, J. R. Laporte, and D. Capella. 2010. "Preliminary efficacy and safety of an oromucosal standardized cannabis extract in chemotherapy-induced nausea and vomiting." Br J Clin Pharmacol 70 (5): 656-63. https://doi.org/10.1111/j.1365-2125.2010.03743.x. http://www.ncbi.nlm.nih.gov/entrez/query.fcgi?cmd=Retrieve&db=PubMed&dopt=Citation&list_uids=21039759.

This one, employing a cannabis-based medicine approved in 30 countries would similarly lend credence to the contention that a cannabis-based pharmaceutical may be applicable to pediatric cancer patients. 

Author Response

Reviewer 2

Comments to the Author:

As important as this topic is, my bottom line conclusion is that this manuscript is simply too preliminary. It may be important to know for whom cannabis has been recommended, but without data on which cancer patients received it, how they benefitted, and what adverse events were associated, there is simply not enough information to merit publication. I strongly encourage the authors to try again when a complete data set is available. 

RESPONSE: There is almost no data looking at the role of medical cannabis in the pediatric and young adult oncology patient therefore we think this paper is important because it describes the demographics of our patients enrolled in the Minnesota Medical Cannabis Registry, including the underlying cancer diagnoses and reason for requesting medical cannabis.  It demonstrates that pediatric and young adult cancer patients are interested in pursuing treatment with medical cannabis for a variety of tumors and symptoms.  We plan to review our charts and interview our patients to determine their compliance with taking medical cannabis and the response of their symptoms to treatment, which we hope to publish at a later date. 

Additional comments:

1) One mention of "marijuana" is sufficient. The proper name is cannabis and "marijuana" has no place in modern scientific discourse.

RESPONSE: We removed marijuana from the manuscript unless the cited study used marijuana.

2) The literature review must be much more intensive and take into account the vast range of available publications. One example is particularly important: 

Abrahamov, A., and R. Mechoulam. 1995. "An efficient new cannabinoid antiemetic in pediatric oncology." Life Sci 56 (23-24): 2097-102.

This type of data showing marked benefit of delta-8-THC on chemotherapy-associated nausea without associated adverse events would greatly bolster the authors' observations.

Another: 

Duran, M., E. Perez, S. Abanades, X. Vidal, C. Saura, M. Majem, E. Arriola, M. Rabanal, A. Pastor, M. Farre, N. Rams, J. R. Laporte, and D. Capella. 2010. "Preliminary efficacy and safety of an oromucosal standardized cannabis extract in chemotherapy-induced nausea and vomiting." Br J Clin Pharmacol 70 (5): 656-63. https://doi.org/10.1111/j.1365-2125.2010.03743.x. http://www.ncbi.nlm.nih.gov/entrez/query.fcgi?cmd=Retrieve&db=PubMed&dopt=Citation&list_uids=21039759.

This one, employing a cannabis-based medicine approved in 30 countries would similarly lend credence to the contention that a cannabis-based pharmaceutical may be applicable to pediatric cancer patients. 

RESPONSE: We added the reference from Abrahamov along with a few other references which are clinical trials that dronabinol and nabilone to be effective for chemotherapy induced nausea and vomiting seen in pediatric oncology patients.  We did not add the reference from Duran because the study only included patients > 18 years of age and the focus of our manuscript is on the pediatric oncology population

Round 2

Reviewer 1 Report

Thank you for the opportunity to review the revised manuscript children-497290 entitled “Medical cannabis certification in a large pediatric oncology center”.

In my opinion, the revised manuscript has improved but I am not sure if the data presented and the conclusions drawn (even after revision) are sufficient for publication. The data presented relate to patient certification and demographic data only. The actual patient characteristics, eg. in terms of their disease (relapsed, BMT, second line therapy etc.) are still missing with the authors pointing to a future publication. These are important points that relate to the demographic data presented.  It remains unclear e.g. why the leukemia patients are so old, it remains unclear at what time point in therapy the certification occurs. In contrast to this missing data, the authors emphasize their philosophy of utilizing cannabis early in the therapy in the discussion, which does not make sense to me. I would like to encourage the authors to fortify their clinical experience/philosophy by adding data and then to combine this with the demographic data presented here.

Author Response

Reviewer 1

Comments to the Author:

Round 2: “In my opinion, the revised manuscript has improved but I am not sure if the data presented and the conclusions drawn (even after revision) are sufficient for publication. The data presented relate to patient certification and demographic data only. The actual patient characteristics, eg. in terms of their disease (relapsed, BMT, second line therapy etc.) are still missing with the authors pointing to a future publication. These are important points that relate to the demographic data presented.  It remains unclear e.g. why the leukemia patients are so old, it remains unclear at what time point in therapy the certification occurs. In contrast to this missing data, the authors emphasize their philosophy of utilizing cannabis early in the therapy in the discussion, which does not make sense to me. I would like to encourage the authors to fortify their clinical experience/philosophy by adding data and then to combine this with the demographic data presented here.”

These are my remarks in detail:

a)     “The data presented relate to patient certification and demographic data only. The actual patient characteristics, eg. in terms of their disease (relapsed, BMT, second line therapy etc.) are still missing with the authors pointing to a future publication.”

RESPONSE: Of note, we added 4 more patients to the original version of the manuscript.  We also added data regarding the number of patients who did not have any medical cannabis dispensed despite being certified in the Minnesota Medical Cannabis Registry and have speculated the reason why they did not follow through on picking up a prescription.  We have also further analysed our data regarding the timing of certification in regards to the actual disease process (i.e. were patients undergoing their first line of therapy or were they being treated for a relapse/tumor progression) and have included this data in our manuscript.

b)     “It remains unclear e.g. why the leukemia patients are so old, it remains unclear at what time point in therapy the certification occurs.”

RESPONSE: We added a comment as to why we think the leukemia patients who were certified for medical cannabis are older than the typical age for pediatric leukemia.  We also included the timing of certification in regards to disease state (i.e. at time of first line therapy or at time of relapse/progression).

c)     “In contrast to this missing data, the authors emphasize their philosophy of utilizing cannabis early in the therapy in the discussion, which does not make sense to me. I would like to encourage the authors to fortify their clinical experience/philosophy by adding data and then to combine this with the demographic data presented here.”

RESPONSE:  We presented data showing that 94% of our patients were certified for medical cannabis early in treatment (i.e. before relapse or tumor progression) and we explained why it is our practice to prescribe early in therapy for our brain tumor and leukemia/lymphoma patients.

Reviewer 2 Report

My initial obejctions remain. At this stage the field needs results, not merely an intention to study the issue.

Author Response

Comments to the Author:

Round 2: “My initial objections remain.”

Of note, their objections during Round 1 were: “At this stage the field needs results, not merely an intention to study the issue. As important as this topic is, my bottom line conclusion is that this manuscript is simply too preliminary. It may be important to know for whom cannabis has been recommended, but without data on which cancer patients received it, how they benefitted, and what adverse events were associated, there is simply not enough information to merit publication. I strongly encourage the authors to try again when a complete data set is available. 

RESPONSE: There is almost no data looking at the role of medical cannabis in pediatric and young adult oncology patients, therefore we think this paper is important because it describes the demographics of our patients who were registered in the Minnesota Medical Cannabis Registry, including the underlying cancer diagnoses and reason for requesting medical cannabis.  It demonstrates that pediatric and young adult cancer patients are interested in pursuing treatment with medical cannabis for a variety of tumors and symptoms. In the current version of the manuscript we have added data on when in their cancer diagnosis (i.e. during the initial therapy or following recurrence or tumor progression) was medical cannabis prescribed and we added the number of patients who had medical cannabis dispensed vs those who did not have it dispensed despite being certified for medical cannabis.  We plan to review our charts and interview our patients to determine their compliance with taking medical cannabis and the response of their symptoms to treatment with cannabis, which we hope to publish at a later date.